# Molecular Characterization of UL50 (dUTPase) Gene of Bovine Herpes Virus 1

**DOI:** 10.3390/ani13162607

**Published:** 2023-08-12

**Authors:** Farzana Shahin, Sohail Raza, Xi Chen, Changmin Hu, Yingyu Chen, Huanchun Chen, Aizhen Guo

**Affiliations:** 1The State Key Laboratory of Agricultural Microbiology, Huazhong Agricultural University, Wuhan 430070, China; f.shaheen64@yahoo.com (F.S.); sohail.raza@uvas.edu.pk (S.R.); chenhch@mail.hzau.edu.cn (H.C.); 2College of Veterinary Medicine, Huazhong Agricultural University, Wuhan 430070, China; chenxi@mail.hzau.edu.cn (X.C.); hcm@mail.hzau.edu.cn (C.H.); chenyingyu@mail.hzau.edu.cn (Y.C.); 3Institute of Microbiology, University of Veterinary and Animal Sciences, Lahore 54000, Pakistan; 4Key Laboratory of Development of Veterinary Diagnostic Products, Ministry of Agriculture, Wuhan 430070, China; 5Hubei International Scientific and Technological Cooperation Base of Veterinary Epidemiology, Huazhong Agricultural University, Wuhan 430070, China

**Keywords:** BoHV-1, dUTPase, polyclonal antiserum, cloned, localized, UL50

## Abstract

**Simple Summary:**

We conducted this study to understand how a virus called *bovine herpes virus-1* (BoHV-1) affects cattle. We focused on a specific gene called UL50, which produces a protein called dUTPase. This protein is important for the virus to replicate in infected cells. We analyzed the UL50 gene and found that it was similar across different herpesviruses. We then cloned and expressed the UL50 gene in a bacterium, purified the resulting protein, and used it to create a special antiserum. Through experiments, we discovered that the UL50 protein was present in infected cells and located in the cytoplasm. These findings help us understand how BoHV-1 causes disease in cattle and may aid in developing control measures.

**Abstract:**

Bovine herpes virus -1 (BoHV-1) infection leads to upper respiratory tract infection, conjunctivitis and genital disorders in cattle. To control BoHV-1, it is important to understand the role of viral proteins in viral infection. BoHV-1 has several gene products to help in viral replication in infected cell. One such gene is deoxyuridine triphosphate nucleotidohydrolase (dUTPase) also known as UL50. In this study, we analyzed the amino acid sequence of UL50 (dUTPase) using bioinformatics tools and found that it was highly conserved among herpesvirus family. Then, it was cloned and expressed in *Escherichia coli* Rosetta (DE3), induced by isopropy1-b-D-thiogalactopyranoside (IPTG) and the recombinant UL50 protein was purified to immunize rabbits for the preparation of polyclonal antiserum. The results indicated that the UL50 gene of BoHV-1 was composed of 978 nucleotides, which encoded 323 amino acids. Western blot analysis revealed that polyclonal sera against UL50 reacted with a band of 34 kDa. Furthermore, immunofluorescence assay showed that UL50 localized in the cytoplasmic area. Taken together, UL50 was successfully cloned, expressed and detected in BoHV-1-infected cells and was localized in the cytoplasm to help in the replication of BoHV-1 in infected cells.

## 1. Introduction

*Herpesviridae* consist of a widely distributed group of large DNA viruses that cause diseases in humans and other vertebrates. It is further divided into three sub-families, *Alpha*-, *Beta-* and *Gammaherpesvirinae,* on the basis of their biological properties, proteins and sequences [1]. Among them, Bovine herpes virus -1 (BoHV-1) is an important pathogen of cattle, which belongs to family *alphaherpesvirdea.* The genomes of *herpesviruses* encode a subset of genes that are homologous among all members of herpesviruses. One such gene encodes for a deoxyuridine triphosphate nucleotidohydrolase (dUTPase). dUTPases belongs to a family of metalloenzymes. The primary function of herpesvirus dUTPases is to catalyze the dephosphorylation of dUTP to dUMP. In addition, they can reduce the dUTP pool and inhibit the incorporation of uracil into newly synthesized DNA by DNA polymerases [2]. Based on their structure and specificity, dUTPases are generally divided into three subgroups: homotrimeric, monomeric, and homodimeric dUTPases. The homotrimeric UTPases is the largest and have high specificity for dUTP. The monomeric dUTPases, which are evolved from the trimeric dUTPases by gene duplication, were found only in herpesviruses [3,4]. The homodimeric dUTPases are different from the monomeric and homotrimeric dUTPases and have wide specificity for dUTP [5]. This enzyme is mostly not necessary for viral replication in vitro [6,7,8] but proved to be essential in vivo [9,10].

PRV *UL50* encodes a 33 kDa dUTPase that is homologous to the dUTPases of other alpha herpesviruses and dispensable for virus replication in infected cells [11], while HSV-1 dUTPase encodes a 40 kDa protein [12]. The mutation of the dUTPase in HSV-1 results in a decrease in its dUTPase activity, which prevents the incorporation of uracil into newly replicating viral DNA, and reduced viral replication and virulence [13]. BoHV-1 also encodes a dUTPase known as *UL50,* comprising 325 amino acids (34 kDa), which is homologous to other sequences in alphaherpesviruses. Previous studies investigating the functions of *UL50* (dUTPase) and some other BoHV-1 enzymes, such as ribonucleotide reductase and DNA polymerase, were aimed to improve the understanding of BoHV-1 pathogenesis and explore their potential application as a useful antiviral target by [14,15,16].

In the current study, we report the identification, cloning and molecular characterization analysis of the BoHV-1 UL50 gene and its prokaryotic expression. The *UL50* protein was purified and used to immunize rabbits for the preparation of polyclonal antibody. Using the rabbit polyclonal antiserum, we examined the intracellular localization of the *UL50* protein and the expression of the *UL50* protein in BoHV-1-infected cells using immunofluorescence assay and Western blotting, respectively.

## 2. Materials and Methods

### 2.1. Ethical Statement

The animal experiment was conducted in Hubei Province, China, and adhered to the guidelines and regulations set forth by the China Hubei Province Science and Technology Department. The ethical considerations were taken into account and approved by the animal ethics committee of Huazhong Agricultural University in Wuhan, China. The research followed the Animal Welfare Regulations for the Administration of Experimental Animals (1988) and the Hubei Regulations for the Administration of Experimental Animals (2005).

### 2.2. Cell Lines

MDBK cells were grown in DMEM growth medium (Hyclone, Logan, UT, USA) supplemented with 10% fetal bovine serum (FBS) (Gibco, Grand Island, NY, USA) and a 2% penicillin and streptomycin (P/S) solution (Hyclone, Linz, Austria, Europe) at 37 °C in a humidified 5% CO_2_ incubator [17,18].

### 2.3. Construction of Expression Vector

For the construction of the expression vector, the entire coding sequence of *UL50* (GenBank accession: AJ004801.1) were amplified from the DNA extracted from BoHV-1 strain IBRV HB06 (isolated by this lab and stored as no. CCTCC V201024 in the Tissue Culture Collection Center of China at Wuhan University) using primers UL50 (F) = 5′GCGGATCC ATGGCAAACAGCGCGGCGGCCACAA’3 and UL50 (R) = 5′CGAAGCTT TTACAGCCCGGTGGACCCAAAGCCG’3 using KOD-Plus (Toyobo, Osaka, Japan). The amplified PCR product was purified using the Cycle Pure Kit (Omega Bio-Tek, Norcross, GA, USA). Subsequently, after enzyme digestion using BamHI and HindIII the product was purified from gel using the TIANgel Midi Purification Kit (Tiangen Biotech, Beijing, China) and confirmed by DNA sequencing by Shanghai Sangon Biological Engineering Technology & Services Co., Ltd. (Shanghai, China). The purified enzyme-digested PCR products were ligated into the prokaryotic expression vector pGEX kg using the GST Gene Fusion System (Novagen, Darmstadt, Germany). The cloned product was confirmed by DNA sequencing (Shanghai, China) and named pGEX-UL50.

### 2.4. Expression and Purification of the GST-Tagged UL50 Fusion Protein

To obtain the highest yield of the fusion protein, the expression plasmids were introduced into *Escherichia coli* Rosetta (*E. coli*) strain BL21 (DE3) (TransGen Biotech, Beijing, China). The expression of the glutathione S-transferase (GST) fusion protein was controlled by an isopropyl-β-D-thiogalactopyranoside (IPTG)-inducible lac operator sequence. For induction, the UL50 bacterial culture was treated with 0.2 mM IPTG and incubated at 20 °C for 16 h. The bacterial cells were harvested by centrifugation (10,000 RPM/10 min), then resuspended in 1× PBS and lysed through sonication. The GST fusion proteins were subsequently eluted from the lysate using Glutathione Sepharose 4B (GE Healthcare, Uppsala, Sweden) and solubilized in SDS loading buffer (containing 10% SDS, 50% glycerol, 0.05% bromophenol blue, 0.5 M Tris-HCl, pH 6.8 and 100 mM DTT). Then, the pellet was boiled at 95 °C for 10 min and centrifuged briefly. The purified eluted protein was analyzed via sodium dodecyl sulfate–polyacrylamide gel electrophoresis (SDS-PAGE) using 12% polyacrylamide gel (Bio-Rad, Hercules, CA, USA).

### 2.5. Rabbit Immunization and Antisera Development

Subsequently, the purified proteins were used to immunize rabbits, as described previously [19]. Briefly, the protocol is described below. Three New Zealand white rabbits, specific-pathogen-free (NZW SPF), 6-to-9 months old, were used for the development of antiserum. For immunization, the purified proteins (0.5 mg) were injected intradermally with Freund’s Complete Adjuvant (FCA) (1:1) or Incomplete Freund’s Adjuvant (IFA). Subsequently, at day 14, 28 (0.5 mg) and 42 (0.1 mg) rabbit were each boosted intravenously. At day 60, the blood serum was collected from veins of the rabbits’ ears and stored at −80 °C for further use.

### 2.6. Western Blot Analysis

To analyze the reactivity and specificity of anti-UL50 antiserum Western blot analysis was performed, as described previously [20]. Briefly, the wild-type BoHV-1 and recombinant fusion protein were separated on 12% (*w/v*) SDS-PAGE (Bio-Rad) and transferred to polyvinylidene fluoride (PVDF) membranes (Millipore, Hong Kong, China). The membrane was blocked in 5% skim-milk-containing PBS-T (0.2% Tween-20 in PBS, PH 7.4) and probed with polyclonal rabbit antisera against *UL50* (1:3000 dilutions) or with a β-actin antibody as an internal reference (1:1000; Beyotime, Haimen, China). Membranes were incubated with secondary antibodies using goat anti-rabbit and goat anti-mouse conjugated with horseradish peroxidase (HRP; 1:5000; Southern Biotech, Birmingham, MI, USA). Blots were detected using chemiluminescent substrate (Thermo Fisher Scientific, Waltham, MA, USA), and images were obtained using film.

### 2.7. Immunofluorescence and Confocal Laser Scanning Microscopy (CLSM)

To examine the subcellular distribution of the UL50 protein, an immunofluorescence assay was conducted. MDBK cells were seeded on 25 mm glass coverslips in 6-well tissue culture plates and either left uninfected or infected with vBoHV1 virus at a multiplicity of infection of 3. At 18 and 24 h post-infection (hpi), the cells were fixed using 4% paraformaldehyde, permeabilized with 0.2% Triton X-100 and then blocked with a solution of 1% bovine serum albumin (Biosharp, Hefei, China) diluted in PBS containing 0.1% Tween 20. Subsequently, the coverslips were incubated with UL50 rabbit polyclonal antisera (diluted 1:100). Finally, the cells were stained with fluorescein isothiocyanate-conjugated goat anti-rabbit IgG antibodies (diluted 1:1000; Beyotime). After each step, the cells were washed three times with 1× PBS. The cell nucleus was counterstained with DAPI (Beyotime), resulting in a blue coloration. The fluorescence was observed using a 152 Zeiss LSM 880 CLSM (Carl Zeiss, Jena, Germany) equipped with a 63× objective.

### 2.8. Bioinformatics Analysis of the UL50 Protein

The amino acid sequence of BoHV-1 UL50 (GenBank accession: AJ004801.1) was retrieved from NCBI (www.ncbi.nlm.nih.gov). The nucleotide sequences aligned and were characterize using Clustal-X program (DNASTAR, Inc., Madison, WI, USA). A similarity in amino acid sequences was evaluated using the basic local alignment search tool (http://www.ncbi.nlm.nih.gov/blast/Blast.cgi (accessed on 25 December 2016)). Phosphorylation sites, glycosylation sites, and the andantigenic peptide in the protein were predicted using http://www.cbs.dtu.dk/services/NetPhos/ (accessed on 25 December 2016), http://www.cbs.dtu.dk/services/NetNGlyc/ (accessed on 25 December 2016) and https://omictools.com/protein-sequence-analysis (accessed on), [21] respectively. The subcellular localization was predicted using Predict NLS program (http://www.rostlab.org/services/predictNLS/ (accessed on 30 December 2016)) and LOCtree program (http://www.rostlab.org/cgi/var/nair/loctree/query (accessed on 30 December 2016)), and signal peptides analysis was performed using SignalP version 3.0 (http://www.cbs.dtu.dk/services/SignalP/ (accessed on 30December 2016) [22,23,24]. The 3D model of the UL50 protein was built using the free online source SWISS-MODEL (https://swissmodel.expasy.org/ (accessed on 20 July 2023).

## 3. Results

### 3.1. Identification and Molecular Characteristics of BoHV-1 UL50 Gene

The open reading frame (ORF) of the BoHV-1 UL50 gene consisted of 978 base pair (bp), consisting of 323 amino acids and potentially encoded a protein of 34.76 kDa, with an isoelectric point of 10.151 as well as a 68.81% guanine/cytosine and 31.19% adenine/thymine ratio. SignalP 4.0 analysis showed that transmembrane and signal peptide regions were absent on the U50 protein. The Netphos-3.1b prediction analysis showed that there are 40 possible potential phosphorylation sites, including serine 18, threonine 15 and tyrosine 5 when the threshold is above 0.5 along the amino acid sequence (Figure 1A). As signal peptides are absent on this protein, it is unable to be exposed to the N-glycosylation machinery even though they contain only potential motifs. NetNGlyc1.0 analysis shows that the BoHV-1 UL50 protein contains eight potential N-linked glycosylation sites at amino acid positions of 3, 24, 69, 71, 195, 222, 245 and 277 when the threshold of prediction score is above 0.5 (Figure 1B). Predicted antigenic peptides analysis determined the presence of thirteen antigenic sits on the surface the *UL50* protein at amino acid position of 16, 28, 58, 76, 95, 109, 160, 179, 194, 235, 247, 282 and 302, with an average antigenic tendency of 1.0365 (Figure 1C). The PredictNLS analysis showed the absence of nuclear localization signal (NLS) on *UL50,* while LOCtree analysis predicted BoHV-1 UL50 localization in the cytoplasm. As previous studies have shown, the *UL50* protein was conserved among the herpesvirus family. The CLUSTAL-W2 software analysis revealed that the *UL50* family members of herpesviruses contain at least eight conserved glycine residues as well as three conserved alanine, two-conserved aspartate, two-conserved serine, two-conserved arginine and one-conserved valine residue in the protein sequences (Figure 2). Furthermore, the sequences retrieved from NCBI with the following GenBank accession nos., BoHV-1 (AJ004801.1), HSV-1 (X14112.1), HSV-2 (NC-001798.2) and PRV (NC-006151.1), were submitted to a BLAST search using the blastn algorithm. The results showed that BoHV-1 UL50 shared identities of 33% with HSV-1, 34% with HSV-2 and 44% with PRV, indicating a potential similar function. The UL50 protein is a medium-sized protein consisting of 323 amino acid residues. It adopts a specific 3D fold essential for its biological function. Among the secondary structural elements observed in the UL50 protein, there are six helices, accounting for approximately 2% of the protein’s sequence. These helices are compact and tightly coiled regions of the polypeptide chain, contributing to the proteins overall stability and shape. Additionally, there are 114 beta strands, making up approximately 46% of the protein’s structure. Furthermore, the UL50 protein contains 124 coil regions, constituting about 50% of its sequence. A notable feature of the UL50 protein is the presence of 60 turns, representing approximately 24% of the protein’s structure. Turns are short segments that reverse the direction of the polypeptide chain and facilitate changes in the protein’s topology. Turns often act as hinges, allowing different parts of the protein to move relative to each other, which can be crucial for its function.

### 3.2. Construction and Cloning of the PGEX-UL50

A complete schematic map of the *UL50* (dUTPases) gene encoding 978 base pairs (bp) in BoHV-1 genome is shown in Figure 3A. The UL50 (dUTPases) gene contains HindIII and BamH1 restriction enzyme sites cloned into pGEX-KG, as shown in Figure 3B. To verify the BoHV-1 UL50 construct, the entire coding sequence of UL50 was amplified using PCR with BoHV-1 UL50 primers from the BoHV-1 strain. A predicted 979 bp DNA product was obtained and confirmed by gel electrophoresis, as shown in Figure 3C. The PCR products were cloned into the prokaryotic expression vector pGEX kg via the BamHI and HindIII restriction sites. The cloned products were confirmed using enzyme digestion (Figure 3D), further confirmed using DNA sequencing and named pGEX-UL50. The sequencing result of UL50 was consistent with the ORF found in the genomic library of BoHV-1. These results showed that the expression vector was successfully constructed.

### 3.3. Purification of the Recombinant BoHV-1 UL50 Protein and Western Blot Analysis to Confirm Specificity of Anti-UL50 Antiserum

The expression plasmid pGEX-UL50 was transformed into *Escherichia coli* (*E.coli*) Rossetta (DE3) to obtain a highly expressed level of *UL50* protein.The glutathione S-transferase [25] fusion protein was expressed in *E. coli* using isopropylthio-beta-D-galactoside (IPTG) for 16 h at 37 °C. SDS-PAGE analysis showed that GST fusion protein was expressed as a soluble form in *E. coli* after induction by IPTG (Figure 4A). The GST fusion proteins were separated from the lysate using slurry Glutathione Sepharose 4B. The eluted purified protein was separated on SDS-PAGE gel. After Coomassie brilliant blue staining, a 63 kDa band for *UL50* was observed because GST possesses a molecular weight of approximately 29 kDa, while the molecular weight of *UL50* is 34 kDa (Figure 4B).

The anti-UL50 polyclonal antiserum was produced in rabbits as described in the Methods section. The reactivity and specificity of anti-UL50 antiserum was tested by Western blot analysis. Polyclonal antisera against GST-UL50 reacted with lysates of the vBoHV1 virus (Figure 4C lane 1) and an IPTG-induced cell (Figure 4C, lane 2) with an apparent molecular mass of 34 kDa. However, no specific band was detected in the cells without induction (Figure 4C, lane 3).

### 3.4. Subcellular Localization of the UL50 Protein in BoHV-1-Infected Cells

To investigate the intracellular localization of the BoHV-1 UL50 protein, we conducted indirect immunofluorescence assays. MDBK cells were either mock-infected or infected with vBoHV-1 virus at a multiplicity of infection (MOI) of 3. Fluorescence imaging was performed using confocal microscopy at 18 and 24 h post-infection (hpi). At 18 hpi, the UL50 protein was primarily detected in the perinuclear cytoplasmic region (Figure 5A), whereas at 24 hpi, the UL50-specific fluorescence was predominantly observed in the cytoplasm (Figure 5B). As expected, no specific fluorescence was observed in the mock-infected cells, which served as internal controls.

## 4. Discussion

The BoHV-1 UL50 gene encodes a dUTPase protein consisting of 323 amino acids. This enzyme is a cellular enzyme and is highly conserved across all herpesviruses. In our study, we focused on the cloning, expression, characterization and localization of the UL50 gene from the BoHV-1 strain. Through the multiple sequence alignment of UL50 (dUTPases) proteins, we observed that BoHV-1 UL50 shares significant similarity with other members of the herpesvirus family. Additionally, the UL50 protein sequence contains several conserved amino acid residues. Notably, it possesses eight conserved glycine residues, three conserved alanine residues, two aspartate residues, two serine residues, two arginine residues, and one valine residue. Previous research has identified five conserved amino acid motifs in dUTPases that are present in all herpesviruses, including BoHV-1. Our comparative analysis of amino acid sequences with other alphaherpesviruses revealed a close relationship between the BoHV-1 UL50 gene product and the corresponding proteins of the pseudorabies virus (PRV), herpes simplex virus type 1 (HSV-1) and herpes simplex virus type 2 (HSV-2) [11,26].

As a crucial initial step in understanding the UL50 protein, we successfully expressed the BoHV-1 UL50 gene in the *E. coli* strain BL21 (DE3). The recombinant UL50 plasmid was carefully verified through both restriction digestion and DNA sequencing. The sequencing analysis confirmed the absence of any nucleotide errors in the synthetic UL50 gene, ensuring its accuracy. To further characterize the UL50 protein, we generated polyclonal antiserum against UL50 by immunizing rabbits with the recombinant UL50 protein. Western blotting analysis using this antiserum revealed the specific recognition of the recombinant UL50 protein. These results indicated that the recombinant UL50 protein induced a strong immunological response, and the UL50 polyclonal antiserum exhibited a high level of specificity.

Furthermore, we investigated the reactivity of the UL50 polyclonal antiserum with BoHV-1-infected cells. Remarkably, the antiserum specifically reacted with BoHV-1-infected cells, demonstrating its ability to detect the UL50 gene product. The apparent molecular mass of the UL50 protein in bovine herpesvirus 1 (BoHV-1) was determined to be approximately 34 kDa, which aligns with the expected size of the UL50 protein. Interestingly, similar observations have been made in other herpesviruses, such as the pseudorabies virus (PRV), in which the UL50 protein was detected at approximately 33 kDa using specific polyclonal antiserum raised against PRV UL50 [11]. Comparing the sizes of dUTPases across different herpesviruses reveals notable distinctions. For instance, the Epstein–Barr virus dUTPase has a molecular mass of approximately 30 kDa [27]. On the other hand, the equine herpesvirus 1 dUTPase is larger, measuring about 36 kDa [28], while the HSV-1 dUTPase is even larger, at approximately 41 kDa [12]. The dUTPase of varicella-zoster virus exhibits an even greater size, measuring around 44 kDa [29]. Herpesvirus dUTPases are notably distinct from dUTPases found in HSV-1 and the varicella-zoster virus due to their significantly larger size. Understanding the differences in size and structural characteristics of dUTPases among herpesviruses can provide valuable insights into their roles and functions in viral replication, nucleotide metabolism and pathogenesis.

These findings highlight the successful expression and characterization of the BoHV-1 UL50 gene product. The production of polyclonal antiserum against UL50 allowed for the specific detection of the protein in Western blotting experiments, indicating its immunogenicity and the high specificity of the generated antiserum. Furthermore, the reactivity of the antiserum with BoHV-1-infected cells further confirms the presence and localization of the UL50 protein during viral infection. The consistency of these observations with PRV UL50 reinforces the significance and conservation of UL50 proteins across related herpesviruses [11]. These results showed that the polyclonal antiserum had a high level of reactivity and specificity.

Furthermore, we used the generated antiserum to investigate the intracellular localization of the BoHV-1 UL50 protein. An indirect immunofluorescence assay combined with confocal microscopy was performed on BoHV-1-infected cells to examine the protein’s localization. Our data indicated that the UL50 protein was predominantly localized in the perinuclear cytoplasmic region and the cytoplasm of infected cells. Notably, we did not observe UL50 localization in the nucleus. Similar findings have been reported for HSV-2, in which UL50 (dUTPase) was found to be localized in the cytoplasm of infected cells [30]. In contrast, previous studies have reported conflicting results for BoHV-1 UL50 (dUTPase), as it was found to be mainly confined to the cytoplasm but was also detectable in the nucleus. In HSV-1, UL50 (dUTPase) was primarily found in the nucleus [26]. The UL50 gene of cytomegalovirus encodes the pUL50 protein and also localizes primarily in the nuclear envelope. There, pUL50 plays a critical role in facilitating the nuclear egress of CMV particles, allowing the virus to spread and establish infection [31]. In pseudorabies virus (PRV), the UL50 gene encodes the pUL50 protein, which is involved in the nuclear egress of viral particles. While the specific localization of pUL50 in PRV has not been extensively studied, it is expected to localize primarily in the nuclear membrane, similar to its localization in related alphaherpesviruses, like HSV-1.

Interestingly, our bioinformatics analysis revealed the absence of a nuclear localization signal (NLS) in the BoHV-1 UL50 protein. The absence of a nuclear localization signal (NLS) in the UL50 protein suggests that it does not possess a specific sequence motif responsible for targeting the protein to the nucleus. NLSs are typically present in proteins that need to enter the nucleus for their proper function. Additionally, protein localization prediction suggested that the UL50 protein is predominantly located in the cytoplasm. This prediction aligns with our immunofluorescence subcellular localization results. However, considering that the replication of all herpesviruses occurs in the nucleus, the implications of these diverse observations remain unknown.

While the focus of this study was on BoHV-1 UL50 localization, it is important to note that UL50 proteins in other herpesviruses may exhibit different intracellular distributions. For instance, UL50 (dUTPase) localization has been reported to be primarily in the cytoplasm of HSV-2-infected cells. In contrast, the UL50 protein of BoHV-1 has been observed to be mainly confined to the cytoplasm but also detectable in the nucleus. Furthermore, in HSV-1, UL50 (dUTPase) was found to be predominantly localized in the nucleus [11].

The localization of the UL50 protein may vary across different viruses within the herpesvirus family. These variations in UL50 localization patterns among different herpesviruses could be attributed to several factors, including differences in viral replication strategies, host cell interactions and the specific functions of UL50 in each virus. The presence or absence of nuclear localization signals (NLS) in the UL50 proteins and interactions with other viral or cellular factors may also play a role in determining their subcellular localization.

Further studies are necessary to explore the underlying mechanisms and functional significance of UL50 localization in different herpesviruses. Comparative analyses across a broader range of herpesviruses will help elucidate the factors influencing UL50 localization and its implications for viral replication and pathogenesis.

Our study provides valuable insights into the intracellular localization of the BoHV-1 UL50 protein. The protein was found to be predominantly localized in the perinuclear cytoplasmic region and cytoplasm of infected cells, similar to findings in HSV-2. The absence of an NLS in the BoHV-1 UL50 protein, along with its cytoplasmic localization prediction, supports our experimental results. However, the underlying reasons for the differential localization patterns observed among different herpesviruses, and the significance of these variations, remain to be elucidated. Further research is needed to unravel the implications of UL50 protein localization and its precise role in BoHV-1 infection. Investigating its interactions with other viral and host proteins, its potential translocation to the nucleus at specific stages and the impact of its localization on viral replication and pathogenesis will contribute to a deeper understanding of BoHV-1 biology.

## 5. Conclusions

In conclusion, we accomplished the successful expression of the BoHV-1 UL50 gene in a prokaryotic expression system. This allowed us to identify and characterize the gene product, which was determined to be a 34 kDa protein. Through immunofluorescence studies, we gained valuable insights into the subcellular localization of UL50 in BoHV-1-infected cells, revealing its predominant localization in the cytoplasm. The successful expression of the BoHV-1 UL50 gene in a prokaryotic system not only provided us with a sufficient amount of the UL50 protein for further analysis but also enabled the characterization of its molecular weight. The determined size of 34 kDa confirmed the expected size of the UL50 protein.

The immunofluorescence studies played a crucial role in elucidating the intracellular localization of UL50 in BoHV-1-infected cells. The observation that UL50 primarily localized in the cytoplasm provides important insights into its cellular distribution during viral infection. This information enhances our understanding of UL50’s potential role in the viral replication cycle and its interactions with other cellular components. Overall, our findings contribute to a better understanding of the BoHV-1 UL50 gene and its gene product, shedding light on its expression, molecular weight and subcellular localization. These findings provide a foundation for further investigations into the functional significance of UL50 in the context of BoHV-1 infection and the broader field of herpesvirus research.

## Figures and Tables

**Figure 1 animals-13-02607-f001:**
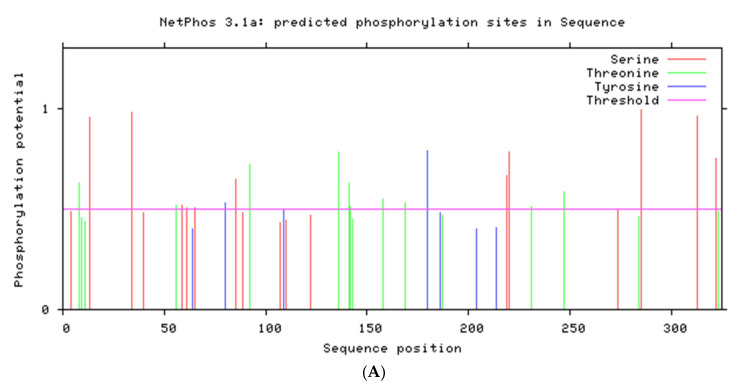
Bioinformatics analysis: (**A**) The prediction result for potential phosphorylation sites of BoHV-1 UL50 protein using NetPhos tool. (**B**) The prediction result for glycosylation sites of BoHV-1 UL50 protein by NetNGlyc1.0. (**C**) Antigenic peptide analysis of BoHV-1 UL50 protein using predicted antigenic peptides tools. (**D**) Three-dimensional structure of UL50 protein.

**Figure 2 animals-13-02607-f002:**
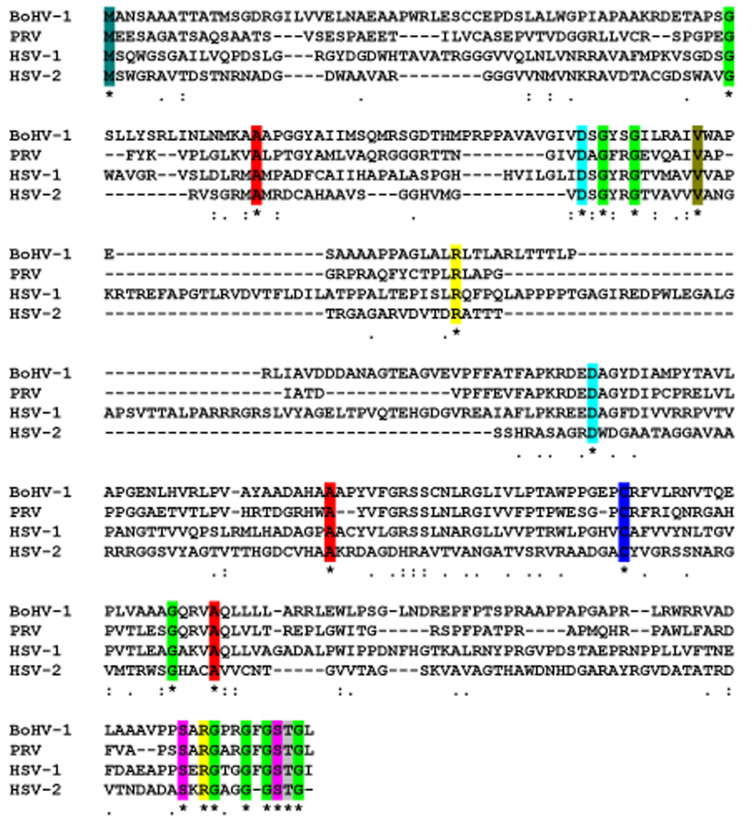
Multiple sequences alignment: comparison between the putative proteins encoded by *UL50* and it homologs, PRV, HSV-1 and HSV-2. Sequences were aligned using the CLUSTALW2 software. Absence of amino acid is shown by dash ‘-’ in the sequences, while ‘*’; ‘:’ and ‘.’ indicate identical amino acid residues, conserved residues and semiconserved residues in all sequence used in the alignment, respectively. Overall conserved residues of M denote with color light green, G green, A red, D light blue, V light brown, R yellow, C blue, S magenta, T gray.

**Figure 3 animals-13-02607-f003:**
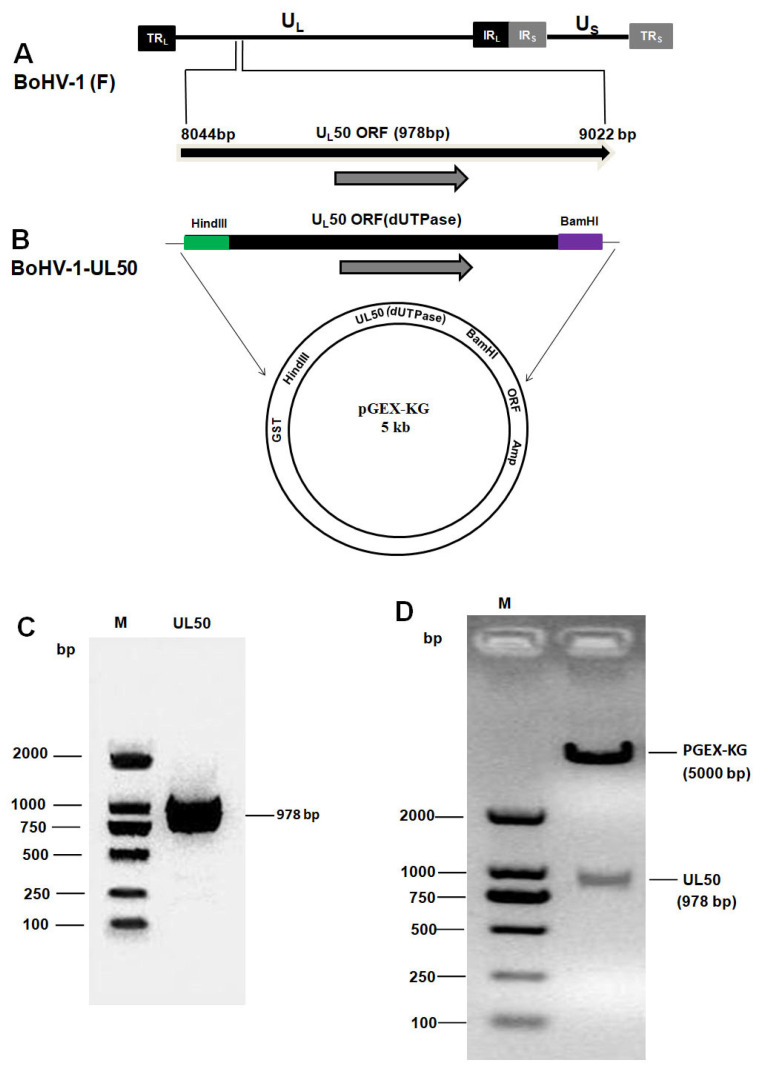
Construction and characterization of BoHV-1 UL50 expression plasmid. (**A**) Map showing the BoHV-1 genome (140 kbp) having unique short (US), unique long (UL), terminal repeat (TR) and internal repeat (IR). Position of *UL50* (dUTPase) located between 8044 bp and 9022 bp on the BoHV-1 genome. (**B**) Cloning of UL50 (dUTPase) containing restriction sites HindIII and BamH1 into vector pGEX-KG. (**C**) Purified UL50 PCR product. (**D**) Enzyme cut with HindIII and BamH1.

**Figure 4 animals-13-02607-f004:**
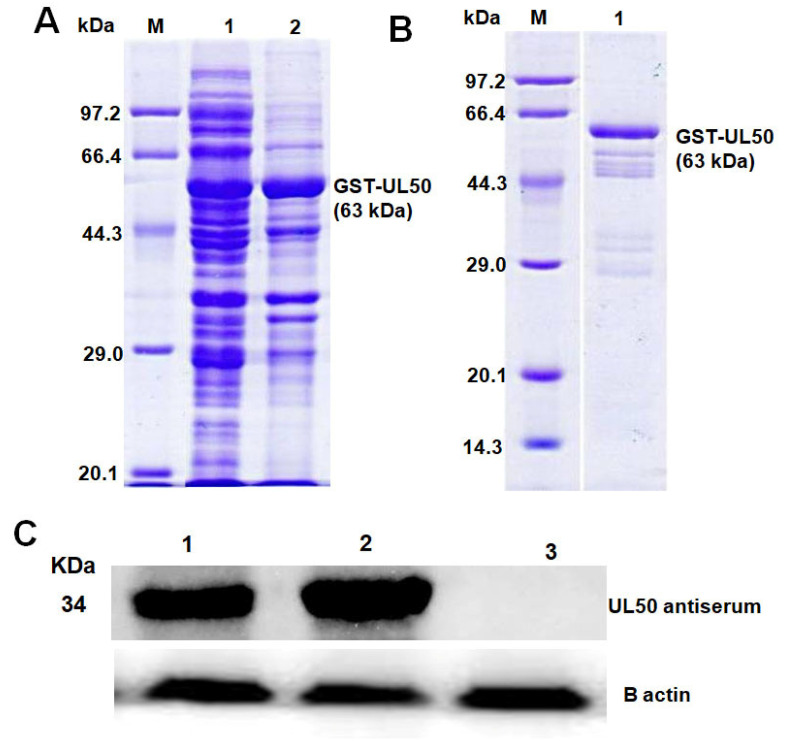
SDS-PAGE of the purified BoHV-1 UL50 and Western blotting analysis. (**A**) Coomassie staining after IPTG protein expression of *UL50* (63 kDa); the GST possess a molecular weight of approximately 29 kDa, while *UL50* has 34 kDa. Supernatant (lane 1) and soluble cell lysate (lane 2), without induction (lane 3), lane M = protein marker. (**B**) UL21 GST-purified protein (lane 1). Flow through (lane 2), lane M = protein marker. (**C**) Western blotting of recombinant protein *UL50* expressed in *Escherichia coli* BL21 (DE3) with the GST-UL50 polyclonal antiserum. GST-UL50 polyclonal antisera reacted with lysates of the vBoHV1 virus (lane 1), IPTG-induced cell (lane 2), cells without induction (lane 3). Beta-actin was used as the internal reference control.

**Figure 5 animals-13-02607-f005:**
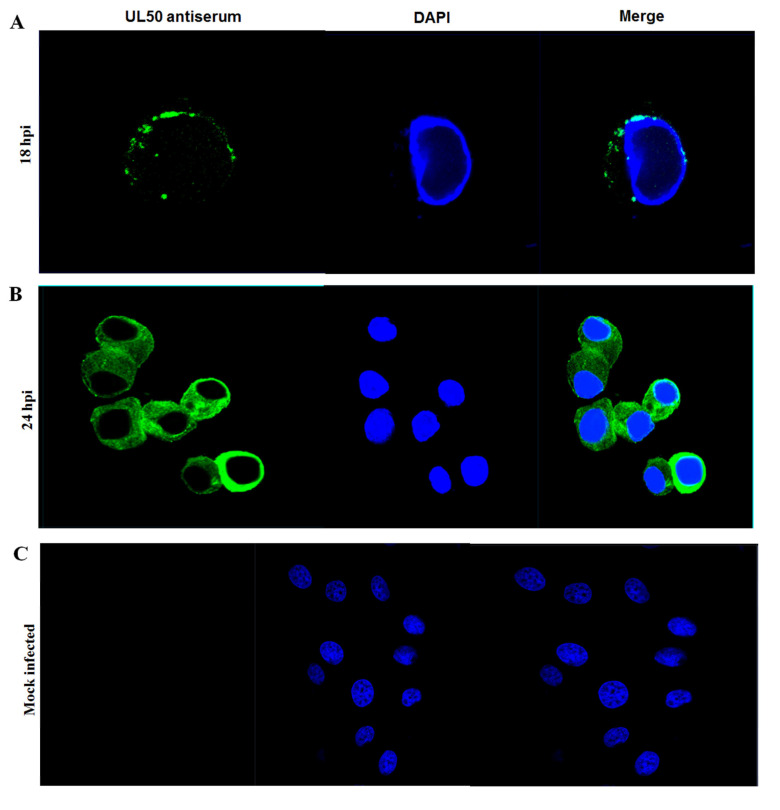
Intracellular localization of the UL50 protein in BoHV-1-infected cells. MDBK cells were mock-infected or infected with BoHV-1 viruses at a MOI 3. The cells were fixed and stained using *UL50* rabbit polyclonal antisera and FITC-conjugated goat anti-rabbit IgG. DNA was stained blue with DAPI at (**A**) 18 hpi (**B**) 24 hpi. (**C**) Mock-infected cells. Fluorescence was observed under a Zeiss LSM 880 laser-scanning confocal microscope (Green: UL50; Blue: Nucleus).

## Data Availability

The data are available upon request from the corresponding author.

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
