# Peer review of "Molecular Characterization of UL50 (dUTPase) Gene of Bovine Herpes Virus 1"

_animals, 2023, doi:10.3390/ani13162607_

Round 1

Reviewer 2 Report

The  authors generated polyclonal rabbit sera directed against dUTPase gene of BoHV-1, showed reactivity by western blotting and immune fluorescence experiments. The manuscript has so many wording, spelling and sentence errors that need correction prior to publication.

In the results and discussion, the authors failed to compare the published findings on dUTPase of other members of herpesviridae to that of BoHV-1 dUTPase. I would recommend the authors to compared other published findings with theirs. dUTPase localization for example

Extensive language and spelling corrections are needed before publication

Reviewer 3 Report

This work cloned and expressed Bovine alphaherpesvirus 1 (BoAHV1) UL50 gene in a prokaryotic system, purified the protein to produce antibodies in rabbits and used confocal microscopy to localize the protein (by immunofluorescence) at 18 and 24 post-infection. The authors also made some bioinformatics analyses of UL50 structure and sequence.

Regarding the characterization of the protein, I would like a more extensive comparison between sequences of UL50 and maybe even a phylogenetic tree. The authors state that it’s a conserved gene in the Order, but only analyzed it with other 3 varicellovirus homologues. I also would have liked if the phosphorylation and glycosylation putative sites (at least some) had been tested.

Although it is new information on BoAHV1 UL50 cell localization, the work is lacking in depth since it is the main focus of the paper. I expected more time points and more information as to the change in localization from 18 hpi to 24 hpi (more time points in between the two presented, especially because of the difference in abundance of the protein in only 6h). Also, the change in magnification between 18 and 24 hpi is a little weird. I would present both fields, one small at the corner and the other as it is (also, it does not say in the legend what magnification was used).

Concerning the writing, firstly the authors need to review herpesvirus taxonomy (there is lot of old nomenclature). Also, the acronym for the virus varies throughout the text. Secondly, I also would like a more detailed and more recent discussion on the knowledge on function and localization of the other herpesvirus dUTPase. Lastly, there is an absurd number of words written together or spaces missing between the period and the next word. Also, genes are written in italic, not proteins.

Now, punctual correction/questionings:

            Line 34 à delete the last 3 “the”

            Line 61 à add PRV complete name.

Missing word at line 71, 81.

Line 84 à change “use” to “used”, “animal” to “animals” and “was” to “were”.

Line 97: are the restriction sites in the primers? If so, please mark it in the sequence.

Line 100 à the product was purified (which size?).

Line 106à product was confirmed.

Line 114 à pellets were

              à change rpm values to x g

              à why was the expression made in a prokaryotic system?

Line 116 à lysed or denatured?

Line 123 à change to “Briefly, three New Zealand…”

Line 125 à Did immunization start on day one or day zero?

Line 127 à each rabbit was

Line 152 à fluorescence was

Line 156 à was aligned

Line 163 à localization was

Line 176 à something wrong with this sentence (so unable?)

Line 190 à delete “having GenBank accession no;”

Line 231 à replace “it” for “its”

Line 237 à gene containing

Line 240 à The predicted

Line 245 à “ in the genomic library of BoAHV1”: which library? I don't understand what did you compare your sequence with. Would it be “consisted with the sequences found in GenBank database”? or “with sequence accession number XXXX”?

Line 289 à UL50 was

Line 293 (and others) à with the “v” in vBoHV1?

Line 299 à supernatant

Line 300 à UL21?????

Line 310 à delete “to”

Line 318 à add magnification

Line 321 à change “and is” to “ but it is”

Line 328 à Is 33-44% closely related?

Line 322 à change “gene” to “protein” (you can’t localize a gene in the cytoplasm)

Line 323 à a BoAHV1

Line 348 à You’re not talking about BoAHV1. Maybe it is PRFV?/ also delete “has been”

Line 349 à delete “to”

            The last paragraph of discussion (lines 356 to 359) should be rewritten and added to the conclusion section.

Reference 25 is missing from the list of references.

Also, the supplemental material should be in English.

Minor editing of English language is required (suggested above).

Supplemental material should be rewritten in English.
